# Periodontitis, Halitosis and Oral-Health-Related Quality of Life—A Cross-Sectional Study

**DOI:** 10.3390/jcm10194415

**Published:** 2021-09-26

**Authors:** Catarina Izidoro, João Botelho, Vanessa Machado, Ana Mafalda Reis, Luís Proença, Ricardo Alves, José João Mendes

**Affiliations:** 1Periodontology Department, Egas Moniz Dental Clinic (EMDC), Egas Moniz—Cooperativa de Ensino Superior, 2829-511 Almada, Portugal; jbotelho@egasmoniz.edu.pt (J.B.); vmachado@egasmoniz.edu.pt (V.M.); ralves@egasmoniz.edu.pt (R.A.); 2Clinical Research Unit (CRU), Egas Moniz Interdisciplinary Research Center (CiiEM), Egas Moniz—Cooperativa de Ensino Superior, 2829-511 Almada, Portugal; jmendes@egasmoniz.edu.pt; 3Neuroradiology Department, Hospital Pedro Hispano, 4464-513 Porto, Portugal; docmaf@sapo.pt; 4ICBAS—Institute of Biomedical Sciences, University of Porto, 4099-030 Porto, Portugal; 5Quantitative Methods for Health Research Unit (MQIS), Egas Moniz—Cooperativa de Ensino Superior, 2829-511 Almada, Portugal; lproenca@egasmoniz.edu.pt

**Keywords:** halitosis, oral-health-related quality of life, periodontitis, periodontal disease, periodontal medicine, volatile sulfurous compounds

## Abstract

We aimed to explore the association between volatile sulfurous compounds (VSCs) and periodontal epithelial surface area (PESA) and periodontal inflamed surface area (PISA) on a cohort of periodontitis patients. Consecutive patients were assessed for periodontitis and halitosis. A full-mouth periodontal status assessment tested probing depth (PD), clinical attachment loss (CAL), gingival recession (REC), bleeding on probing (BoP), PISA and PESA. A halitosis assessment was made using a VSC detector device. Periodontal measures were regressed across VSC values using adjusted multivariate linear analysis. From a total of seventy-two patients (37 females/35 males), the PESA of posterior-lower regions was found to be significantly higher in halitosis cases than their non-halitosis counterparts (*p* = 0.031). Considering all patients, the PESA of the posterior-lower region (B = 1.3, 95% CI: 0.2–2.3, *p* = 0.026) and age (B = −1.6, 95% CI: −3.1–0.2, *p* = 0.026) showed significant association with VSCs. In halitosis patients, the PESA of the posterior-lower region (B = 0.1, 95% CI: 0.0–0.1, *p* = 0.001), PISA Total (B = −0.1, 95% CI: −0.1–0.0, *p* = 0.008) and the OHIP-14 domain of physical disability (B = −2.1, 95% CI:−4.1–0.1, *p* = 0.040) were the most significant variables in this model. The PESA from the posterior-lower region may be associated with VSCs when other causes of extra-oral halitosis are excluded. Further intervention studies are needed to confirm this association.

## 1. Introduction

Periodontitis is a plaque-induced chronic inflammatory disease affecting the periodontium. The destruction of the periodontium is the most common cause of tooth loss worldwide [1]. In a representative Portuguese study, the prevalence of periodontitis was estimated at 59.9%, with 24.0% and 22.2% of the participants exhibiting severe and moderate periodontitis, respectively [2]. The clinical manifestation of periodontitis may include tooth mobility, gingival bleeding, halitosis, masticatory impairment, pain and, ultimately, tooth loss [3], with a negative impact on oral-health-related quality of life (OHRQoL) [4] particularly associated with halitosis [5].

Halitosis results from an unpleasant odor arising from the oral cavity when breathing or speaking. Several lines of evidence estimate that 80–90% of the causes of bad breath arise from the oral cavity [6,7,8] while the remaining causes are due to conditions involving the gastrointestinal tract, the upper and lower respiratory system, the use of medications, diabetes mellitus, liver cirrhosis, uremia and idiopathic conditions [9,10]. Oral malodor is primarily caused by the microbial degradation of both sulfur-containing and non-sulfur-containing amino acids derived from proteins in exfoliated human epithelial cells and white blood cell debris or present in plaque, saliva, blood and tongue coatings [7,11]. Subgingival periodontal biofilm is mainly composed of Gram-negative anaerobic bacterial species, which present a proteolytic nature [12]. Those species are able to degrade sulfur-containing substrates on different surfaces of the oral cavity, including the periodontal pockets, releasing volatile sulfur compounds (VSCs).

Clinical studies have shown that VSCs are the major contributors to halitosis [13,14]. Hydrogen sulfide, methyl mercaptan and, to a lesser extent, dimethyl sulfide represent 90% of the VSC in bad breath [9,13,14]. From a patient’s point of view, halitosis negatively impacts oral-health-related quality of life (OHRQoL), specifically interpersonal relationships, which is the main reason for seeking professional care [15].

Individuals with periodontitis present a higher likelihood of having oral malodor [16]. Moreover, the association between periodontitis and halitosis is still far from being fully understood because evidence comes from studies with different periodontitis criteria and halitosis measurement techniques [17]. Additionally, two periodontal research measures have been recently presented, periodontal epithelial surface area (PESA) and periodontal inflamed surface area (PISA) [18], that more accurately represent the periodontal status of the patient; these measures have never been explored with the quantification of VSCs.

Therefore, the main aim of this cross-sectional study was to explore the association between VSCs and PISA and PESA on a cohort of periodontitis patients. We further assessed whether measures of halitosis (VSCs and self-reported) were associated with OHRQoL.

## 2. Materials and Methods

### 2.1. Study Design and Setting

This cross-sectional study was conducted at the Egas Moniz Dental Clinic (EMDC). Participants were consecutively recruited from the Periodontology Department of EMDC for periodontal assessment between October 2019 and March 2021. This study was approved by the Egas Moniz Ethics Committee (no. 781) in accordance with the Helsinki Declaration of 1975, as revised in 2013. After a full explanation, patients agreed to participate in the study and signed a written consent form.

This study was carried out following the Strengthening the Reporting of Observational Studies in Epidemiology (STROBE) statement [19].

### 2.2. Participants and Eligibility Criteria

We enrolled participants with the following inclusion criteria: had received a diagnosis of periodontitis; were aged between 18 and 65; fulfilled the recommendations given for halitosis assessment; and had given their informed consent. Patients were excluded if they had previously been treated for periodontitis; had consumed antibiotics within the last 4 weeks; had a past history of radiotherapy of the head and neck; had received chemotherapy (previous 6 months); experienced extra-oral halitosis (such as sinusitis, bronchitis, rhinitis, pharyngitis, laryngitis, cancer, diabetes mellitus, kidney diseases and antidepressants drugs) [9,10]; were pregnant; were receiving systemic medication resulting in hyposalivation; or were diagnosed with diabetes mellitus.

Participants were unaware that a halitosis assessment would be performed (to reduce any bias) and were instructed prior to the periodontal assessment to: avoid spicy foods and foods such as onions and garlic (for 24 to 48 h before examination); to avoid smoking 4 to 12 h before the exam; to perform oral hygiene 12 h before the exam if it was to be performed in the morning or 4 h before the exam if it was to be performed in the afternoon; to consume water until 1 h before treatment; and to avoid the use of perfumes and deodorants within 24 h before the test [20].

### 2.3. Variables

#### 2.3.1. Sociodemographic and Medical Questionnaires

The sociodemographic data collected included gender, age, educational level (no education, elementary, middle or higher), occupation status (student, employed, unemployed or retired), marital status (single, married, divorced or widowed), smoking habits (quantity and duration), alcoholic habits (quantity and frequency) and average family monthly income (in EUR). In the medical questionnaire, patients reported the presence of systemic diseases and medications and oral hygiene habits (frequency and devices used). The general dental examination evaluated the presence of: caries, retained roots, fixed prosthesis, removable prosthesis, poorly adapted restorations, supragingival calculus, implants, peri-implantitis, recent extractions and dental abscesses).

#### 2.3.2. Periodontal Assessment

A full-mouth periodontal examination was conducted to assess the following clinical parameters at six sites per tooth using a manual periodontal CP-12 probe (Hu-Friedy^®^, Chicago, IL, USA): probing pocket depth (PPD), clinical attachment loss (CAL) and bleeding on probing (BOP). Plaque index (PI) [21], gingival recession (REC), probing depth (PD) and bleeding on probing (BoP) were circumferentially recorded at six sites per tooth (mesiobuccal, buccal, distobuccal, mesiolingual, lingual and distolingual). PD was measured as the distance from the free gingival margin to the bottom of the pocket and REC as the distance from the cementoenamel junction (CEJ) to the free gingival margin, and this assessment was assigned a negative sign if the gingival margin was located coronally to the CEJ. CAL was calculated as the algebraic sum of REC and PD measurements for each site. The measurements were rounded to the lowest whole millimeter. Furcation involvement (FI) was assessed using a Naber probe [22]. Tooth mobility was further appraised [23]. Periodontitis cases were defined according to the AAP/EFP 2018 consensus, with a patient being a periodontitis case if interdental CAL is detectable at ≥2 mm non-adjacent teeth or buccal or oral CAL ≥3 mm with pocketing >3 mm is detectable at ≥2 teeth [24].

#### 2.3.3. Halitosis Assessment

The diagnosis of halitosis was carried out in two steps: (1) a self-reported questionnaire to exclude possible causes for extra-oral halitosis (referred in Section 2.2) [9,10]; (2) a halitosis assessment through a VSC-monitoring device (Halimeter^®^, Interscan Corp, Chatsworth, CA, USA).

Prior to the assessment, the mouth remained closed for 1 min. The end of the cannula was then inserted into the patient’s mouth, and the VSCs score recorded at the maximum peak displayed by the device. The result was interpreted according to the manufacturer’s instructions, and as previously reported: less than 80 ppb denoted no perceptible odor, 80 to 100 ppb denoted perceivable odor, 100 to 120 ppb denoted moderate halitosis, 120 to 150 ppb denoted more pronounced halitosis and >150 ppb denoted severe halitosis [25,26].

#### 2.3.4. OHRQoLQuestionnaire

In order to assess oral-health-related quality of life (OHRQoL), we applied the Oral Health Impact Profile Questionnaire (OHIP-14) [27] validated for Portuguese language [28]. Answers were given on a Likert scale ranging from 0 to 4 (0 = never, 1 = almost never, 2 = occasionally, 3 = quite often, 4 = very often), and the survey was completed by the patient alone without interference. In cases where five or more items were missing or more than two items were missing in a subscale, the questionnaire was considered invalid; however, all responses were answered in full and considered for statistical analysis [29].

### 2.4. Measurement Reliability and Reproducibility

The clinical data collection was performed by a single examiner previously subjected to a calibration process (intra- and inter-examiner) for the periodontal assessment in five patients not included in the study. Measurement reliability and reproducibility were assessed by the intra-class correlation coefficient (ICC). Inter-examiner agreement was 0.97 for both PD and CAL, and intra-examiner agreement was 0.94 for both PD and CAL.

### 2.5. Statistical Analysis

Data analysis was performed using SPSS for Windows (IBM SPSS Statistics version 26.0 for Windows, IBM Corporation, Armonk, NY, USA). Descriptive and inferential statistics methodologies were applied. Mean values and standard deviation (SD) were calculated for continuous values. For analysis purposes, OHRQoL scores were considered as continuous variables. The explicit comparison of mean values was not performed by parametric tests since data assumptions for the application of the tests were not met (normality and homoscedasticity). A group data comparison was alternatively performed by Mann–Whitney and Kruskal–Wallis tests. A chi-square test was used for comparisons of categorical variables across the groups. The level of significance was set at 5% in all inferential analyses. All patients completed the questionnaires, and so missing data management was not required.

## 3. Results

### Participants

From an initial sample of 132 patients referred for periodontal diagnosis, 60 participants were excluded when inclusion and exclusion criteria were applied (Figure 1).

Thus, a final sample of 72 participants was obtained. Overall, this sample had 48.6% males (*n* = 35) with a mean age of 54.5 (±17.6) years (Table 1). A total of 27.8% were smokers (*n* = 20), and 16.7% and 5.6% had reported cardiovascular disease and asthma, respectively.

Of the 72 participants, 44.4% (*n* = 32) had halitosis, of which 53.1% (*n* = 17) were female. With regard to smoking habits, we observed that 78.1% (*n* = 25) of participants with halitosis were non-smokers and 21.9% (*n* = 7) smokers. Halitosis was not found to be significantly associated with gender (*p* = 0.817) and smoking habits (*p* = 0.429).Regarding oral health conditions, the use of removable prosthesis (*p* = 0.012) and the presence of implants (*p* = 0.035) were significantly more prevalent in halitosis cases.

When analyzing periodontal clinical characteristics according to the halitosis status (Table 2),we found no differences for tooth brushing frequency, interdental cleaning, periodontal staging/extent and most periodontal clinical measures (*p* > 0.05). However, the PESA from posterior-lower regions was found to be significantly higher in halitosis cases than in their non-halitosis counterparts (*p* = 0.031).

As with the OHIP total score and all seven OHIP-14 domains (Table 3), no significant differences were found regarding the halitosis assessment. When we compared different periodontitis stages, a difference was observed regarding physical pain (*p* = 0.048).

We then assessed which variables were associated with the measurement of VSCs both in the overall sample and in a sample of patients clinically diagnosed with halitosis (Table 4). Considering all patients, our stepwise linear regression pointed to the PESA of the posterior-lower region (B = 1.3, 95% CI: 0.2–2.3, *p* = 0.026) and age (B = −1.6, 95% CI: −3.1–0.2, *p* = 0.026) as the most significant variables. As with halitosis patients, the PESA of the posterior-lower region (B = 0.1, 95% CI: 0.0–0.1, *p* = 0.001), PISA Total (B = −0.1, 95% CI: −0.1—0.0, *p* = 0.008) and the OHIP-14 domain of physical disability (B = −2.1, 95% CI: −4.1—0.1, *p* = 0.040) were the most significant variables within this model. A graphical representation of PESA posterior-lower teeth and age towards the measurement of VSCs was made (Figure 2).

## 4. Discussion

In this cross-sectional study, we sought to explore the relationship of PISA and PESA to VSCs. We predicted that exploring such far-reaching measures would provide a more comprehensive understanding on how periodontal destruction relates to VSCs. Our results confirmed that VSC counts may be associated with the amount of PESA of the posterior-lower region. To the best of our knowledge, this study may be the first to demonstrate such an association.

Halitosis occurs via the complex interaction between bacterial biofilm and protein substrates [12]. A subgingival biofilm is composed mainly of anaerobic Gram-negative bacteria of a proteolytic nature, capable of degrading the substrates present in the oral cavity to produce VSCs [16]. The levels of VSC have been shown to increase with periodontitis severity [30]. Biologically, deeper periodontal crevices emerge with the progression of periodontitis, increasing the area of bacterial colonization and intensifying the release of VSCs [31]. Additionally, the hypoxic milieu of deepest pockets, acidification and the consequent activation of the decarboxylation of amino acids give rise to VSCs [31]. In turn, VSCs shift the permeability of the oral mucosa and the solubility of collagen; they also decrease the synthesis of proteins and collagen, resulting in the destruction of periodontal tissues [31,32,33].

The significance of the posterior-lower region may be seen as innovative and shall be clarified in future and prospective studies. According to these results, it may be possible, in the future, to establish a protocol for the treatment of periodontitis and halitosis. Starting periodontal treatment at the posterior-lower sites/regions in order to motivate patients and respond to the patient’s complaints may be effective. Conventional periodontal therapy (scaling and root planning) helps to mitigate Gram-negative anaerobic bacteria, consequently decreasing the concentration of VSCs in the oral cavity and, with this, the prevalence of halitosis. This decrease in VSCs produces a significant improvement in halitosis. The treatment of halitosis can be used by clinicians as a motivating instrument for the patient and for periodontal treatment [15,34,35]. Taking inspiration from the fact that halitosis may affect quality of life, our results revealed no significant differences among the various domains of OHIP-14 according to the halitosis status of patients. This may relate to the self-perception of bad breath, as these participants manifested a clear inability to judge this, as measured through self-perception of bad breath between patients with and without halitosis. Several studies have addressed self-perceived halitosis (SPH), most of them in dental students [33,34,35,36,37]. Inadequate oral hygiene and infrequent tooth brushing [38,39] and unawareness towards halitosis prevention [38] were among the most important related factors. In addition, awareness and concern regarding halitosis have resulted in better extra-oral self-care practices [34,40].

Thus, greater public awareness and education should be encouraged. Informing and educating patients for oral malodor should be emphasized and new generations should be qualified to address this matter effectively [41].

### Strengths and Limitations

This study presents a convenience sample from a Periodontology department, hindering the potential for generalization. Selection bias related to such narrow inclusion and exclusion criteria, as well as difficulties in measuring and diagnosing halitosis, may also have occurred. Thus, new studies in different and larger populations can help to validate our findings. Similarly, future randomized clinical trials are needed to measure the biological and biochemical differences in periodontal and halitosis assessment before and after periodontal treatment and its impact on oral-health-related quality of life.

Halitosis assessment was based on the measurement of VSCs and could therefore have been complemented by organoleptic assessment [16] as VSCs have low sensitivity [36]. Nevertheless, these results were significant even though the association between periodontitis and halitosis is greater in studies where organoleptic examination is carried out [16]. However, it is important to highlight the subjectivity of the organoleptic test [26], despite this test remaining the “gold standard” diagnostic method [37]. In our view, measuring VSC levels constitutes an objective method [8] and, therefore, has higher potential clinical applicability.

The main advantages of the present study can be seen in its design, which differs from other studies by its definition of cases of periodontitis, where the most up-to-date case definition was considered [42], ensuring comparability with future studies. The exclusion of causes of extra-oral halitosis is also a strength. The assessment of tridimensional periodontal measures (PISA and PESA) provides new results that deserve further attention.

## 5. Conclusions

Within the limitations of this observational study, the PESA from the posterior-lower region seems to be associated with VSC levels when other causes of extra-oral halitosis are excluded. Further intervention studies are needed to verify a possible causal association.

## Figures and Tables

**Figure 1 jcm-10-04415-f001:**
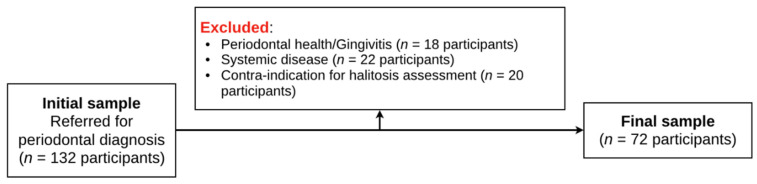
Flowchart of participants.

**Figure 2 jcm-10-04415-f002:**
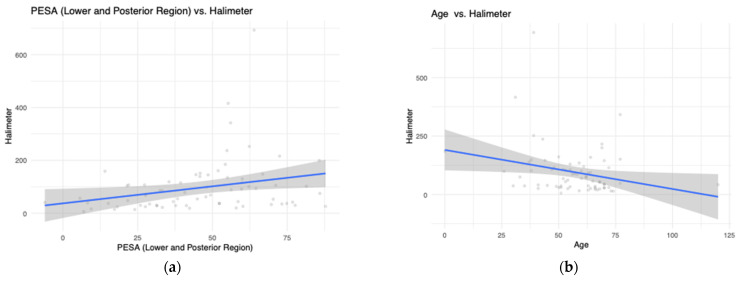
Graphical representation of PESA posterior-lower teeth and age towards the measurement of VSCs. (**a**) PESA (Lower and Posterior Region) vs Halimeter; (**b**) Age vs Halimeter; (**c**) Age*PESA (Lower and Posterior Region) vs Halimeter.

**Table 1 jcm-10-04415-t001:** Participants’ characteristics, stratified according to halitosis status.

	No Halitosis (*n* = 40)	Halitosis (*n* = 32)	*p*-Value *	Total (*n* = 72)
Age (years), mean (±SD)	59.5 (±3.6)	49.9 (±3.2)	0.066	54.6 (±17.6)
Gender, *n* (%)				
Male	20 (50.0)	15 (46.9)	0.817	35 (48.6)
Female	20 (50.0)	17 (53.1)		37 (51.4)
Smoking habits, *n* (%)				
Current smoker	13 (32.5)	7 (21.9)	0.429	20 (27.8)
Non-smoker	27 (67.5)	25 (78.1)		52 (72.2)
Comorbidities, *n* (%)				
Cardiovascular disease	10 (25.0)	7 (21.9)	0.788	12 (16.7)
Asthma	0 (0.0)	4 (12.5)	**0.035**	4 (5.6)
Self-perceived halitosis, mean (±SD)	4.0 (2.6)	4.4 (2.6)	0.567	4.2 (2.6)
Oral health conditions, *n* (%)				
Poorly adapted restorations	6 (15.0)	7 (21.9)	0.543	13 (18.1)
Fixed prosthesis	6 (15.0)	7 (21.9)	1.000	13 (18.1)
Removable prosthetic	17 (42.5)	5 (15.6)	**0.012**	22 (30.6)
Caries	21 (52.5)	13 (40.6)	0.350	34 (47.9)
Retained roots	6 (15.0)	5 (15.6)	1.000	11 (15.3)
Supragingival calculus	30 (75.0)	25 (78.1)	0.788	55 (76.4)
Peri-implantitis	0 (0.0)	3 (9.4)	0.083	3 (4.2)
Presence of implants	0 (0.0)	4 (12.5)	**0.035**	4 (5.6)
Recent extraction	3 (7.5)	2 (6.3)	1.000	5 (6.9)
Presence of dental abscesses	0 (0.0)	2 (6.3)	0.194	2 (2.8)
VSCs, mean (SD)	38.5 (18.0)	168.1 (121.3)	**<0.001**	96.1 (103.9)

* Chi-square test for categorical variables and Mann–Whitney test for continuous variables, *p* < 0.05 denoted in bold.

**Table 2 jcm-10-04415-t002:** Periodontal clinical characteristics, stratified according to halitosis status.

	No Halitosis (*n* = 40)	Halitosis (*n* = 32)	*p*-Value *	Total (*n* = 72)
Tooth brushing frequency, *n* (%)				
Once a day or less	5 (12.5)	7 (21.9)	0.076	12 (16.7)
Twice a day	19 (47.5)	7 (21.9)		26 (36.1)
More than twice a day	16 (40.0)	18 (56.2)		34 (47.2)
Interdental cleaning, *n* (%)				
No	22 (55.0)	15 (46.7)	0.636	26 (51.0)
Yes	18 (45.0)	17 (53.3)		25 (49.0)
Periodontal staging, *n* (%)				
I/II	2 (5.0)	4 (12.5)	0.191	6 (8.3)
III	15 (37.5)	16 (50.0)		31 (43.1)
IV	23 (57.5)	12 (37.5)		35 (48.6)
Periodontal extent, *n* (%)				
Localized	13 (32.5)	11 (34.4)	1.000	24 (33.3)
Generalized	27 (67.5)	21 (65.6)		48 (66.7)
Periodontal clinical parameters, mean (SD)				
BOP (%)	20.9 (18.4)	19.9 (18.3)	0.865	20.5 (18.2)
PI (%)	35.5 (21.5)	30.6 (18.0)	0.371	33.3 (20.0)
CAL (mm)	6.0 (±0.08)	6.0 (±0.05)	0.562	6.0 (0.07)
PPD (mm)	3.1 (±0.8)	3.2 (±0.7)	0.493	3.2 (0.7)
Recession (mm)	1.0 (0.9)	1.0 (1.1)	0.504	1.0 (0.9)
Missing teeth	10.1 (5.7)	7.8 (5.1)	0.066	9.1 (5.5)
PISA Total	89.9 (120.0)	95.3 (107.5)	0.227	92.3 (113.9)
PESA Total	311.8 (179.7)	344.3 (151.5)	0.230	326.3 (167.4)
PISA Posterior	69.8 (102.9)	76.7 (94.7)	0.200	72.9 (98.7)
PISA Anterior	20.1 (22.7)	18.6 (19.9)	0.977	19.4 (21.4)
PISA Posterior-Upper	60.2 (93.4)	65.8 (86.9)	0.268	62.7 (89.9)
PISA Anterior-Upper	7.8 (11.0)	8.5 (14.0)	0.964	8.1 (12.4)
PISA Posterior-Lower	9.6 (13.0)	11.0 (11.7)	0.283	10.2 (12.4)
PISA Anterior-Lower	12.3 (14.1)	10.1 (11.4)	0.623	11.3 (12.9)
PESA Posterior	224.8 (158.2)	261.2 (141.2)	0.239	241.0 (150.9)
PESA Anterior	88.6 (32.0)	94.2 (20.3)	0.272	91.1 (27.4)
PESA Posterior-Upper	184.0 (143.0)	209.7 (134.8)	0.344	195.4 (139.0)
PESA Anterior-Upper	36.9 (21.0)	42.0 (17.2)	0.106	39.1 (19.5)
PESA Posterior-Lower	40.8 (23.9)	51.6 (17.0)	**0.031**	45.6 (21.7)
PESA Anterior-Lower	51.8 (17.4)	52.2 (16.7)	0.721	52.0 (16.9)

* Chi-square test for categorical variables and Mann–Whitney test for continuous variables, *p* < 0.05 denoted in bold.

**Table 3 jcm-10-04415-t003:** Oral-health-related quality of life (OHRQoL) according to halitosis status and periodontal status, presented as a mean (standard deviation) of OHIP-14 domain scores.

OHIP-14 Domains	Halitosis Assessment			Periodontitis Stage				Total (*n* = 72)
	No Halitosis (*n* = 40)	Halitosis (*n* = 32)	*p*-Value *	I/II (*n* = 6)	III (*n* = 31)	IV (*n* = 35)	*p*-Value *	
Functional Limitation	1.58 (1.85)	1.66 (2.42)	0.502	2.67 (2.16)	1.06 (1.57)	1.91 (2.42)	0.214	1.61 (2.11)
Physical Pain	2.88 (2.26)	3.16 (2.45)	0.709	4.67 (2.07)	2.39 (2.11)	3.26 (2.43)	**0.048**	3.00 (2.33)
Psychological Discomfort	3.75 (2.63)	3.94 (2.49)	0.744	3.83 (1.60)	3.39 (2.40)	4.23 (2.79)	0.458	3.83 (2.55)
Physical Disability	2.50 (2.55)	1.97 (2.46)	0.321	3.00 (2.19)	1.55 (1.86)	2.77 (2.92)	0.184	2.26 (2.51)
Psychological Disability	2.30 (2.33)	2.25 (2.16)	0.995	2.83 (2.04)	1.55 (1.46)	2.83 (2.67)	0.129	2.28 (2.24)
Social Disability	1.33 (2.02)	1.28 (1.84)	0.889	1.33 (1.63)	0.74 (1.18)	1.80 (2.36)	0.269	1.31 (1.93)
Handicap	1.85 (2.21)	1.84 (2.05)	0.822	2.00 (2.10)	1.26 (1.61)	2.34 (2.44)	0.202	1.85 (2.13)
Total	16.18 (13.48)	16.19 (12.26)	0.843	20.33 (10.58)	12.00 (7.88)	19.17 (15.69)	0.139	16.18 (12.87)

* Mann–Whitney test. *p* < 0.05 denoted in bold.

**Table 4 jcm-10-04415-t004:** Stepwise multivariate linear regression analysis of age and PESA posterior-lower teeth for the outcome variable of VSCs (*n* = 405).

	Overall (*n* = 72)			Halitosis Patients (*n* = 32)		
Variable	B (SE)	95% CI	*p*-Value	B (SE)	95% CI	*p*-Value
Constant	131.2 (49.4)	32.8; 229.7	**0.010**	36.8 (5.1)	26.4; 47.1	**<0.001**
PESA Posterior-Lower	1.3	0.2; 2.3	**0.021**	0.1	0.0; 0.1	**0.001**
Age	−1.6	−3.1; −0.2	**0.026**	-	-	-
PISA Total	-	-	-	−0.1	−0.1; −0.0	**0.008**
Physical Disability	-	-	-	−2.1	−4.1; −0.1	**0.040**

*p* < 0.05 denoted in bold.

## Data Availability

Data may be available upon reasonable request.

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
