# Peer review of "Periodontitis, Halitosis and Oral-Health-Related Quality of Life—A Cross-Sectional Study"

_jcm, 2021, doi:10.3390/jcm10194415_

Round 1

Reviewer 1 Report

Dear Authors,

The article: Periodontitis, halitosis and oral health-related quality of life – a cross-sectional study was  to explore the association between VSCs with PISA and PESA on a cohort of periodontitis patients. We further assessed whether measures of halitosis (VSCs and self-reported) were associated with OHRQoL.

Abstract should be unstructurized.

Punctuation and editorial errors in the text should be corrected.

line 144 slip to another page.

Figure 2 - onclude figure with better quality.

Incorrect citation record type at reference point.
e.g.: Zhang, J.; Yu, X.; Guo, P.; Firrman, J.; Pouchnik, D.; Diao, Y.; Samulski, R.J.; Xiao, W. Satellite Subgenomic Particles Are Key Regulators of Adeno-Associated Virus Life Cycle. Viruses 202113, 1185.

To sum up, article can be accepted after minor revision.

Reviewer 2 Report

References from 43 until 50 needs reconsiderations!
Very interesting topic, sample size is a bit tiny. I suggest revising the references, and also having a language editing service.

Reviewer 3 Report

Dear Authors, your work is well conducted and very precise. As clinician periodontist, after reading the article, I asked to myself : will this finding improve my tools for motivation?

I believe that this is a crucial stage with periodontally compromised patients! In details reading table 2 in the results section it can be perceived that for ex. halitosis was detected with similar value in different patients typologies ( who do or do not brush regularly their teeth). Moreover, reported BOP values are 20.9 and 19.9 in patients without and with  halitosis respectively; this last consideration may arise a further question " if the halitosis is not directly related to the clinical evolution of the periodontal disease , how can I use it to motivate a patient?".

To conclude, I believe that it a very good exercise of collecting data for a statistical analysis but it may not improve significantly the clinical tools for the everyday practice.

Round 2

Reviewer 3 Report

Dear Authors, thank you for providing answers to my objections. As I said before this research is conducted properly and precisely but it is still  far away from clinical application.